# Apitherapy in Post-Ischemic Brain Neurodegeneration of Alzheimer’s Disease Proteinopathy: Focus on Honey and Its Flavonoids and Phenolic Acids

**DOI:** 10.3390/molecules28155624

**Published:** 2023-07-25

**Authors:** Ryszard Pluta, Barbara Miziak, Stanisław J. Czuczwar

**Affiliations:** Department of Pathophysiology, Medical University of Lublin, 20-090 Lublin, Poland; barbara.miziak@umlub.pl (B.M.); stanislaw.czuczwar@umlub.pl (S.J.C.)

**Keywords:** brain ischemia, Alzheimer’s disease proteinopathy, apitherapy, honey, excitotoxicity, neuroinflammation, apoptosis, amyloid, tau protein, neurodegeneration

## Abstract

Neurodegeneration of the brain after ischemia is a major cause of severe, long-term disability, dementia, and mortality, which is a global problem. These phenomena are attributed to excitotoxicity, changes in the blood–brain barrier, neuroinflammation, oxidative stress, vasoconstriction, cerebral amyloid angiopathy, amyloid plaques, neurofibrillary tangles, and ultimately neuronal death. In addition, genetic factors such as post-ischemic changes in genetic programming in the expression of amyloid protein precursor, β-secretase, presenilin-1 and -2, and tau protein play an important role in the irreversible progression of post-ischemic neurodegeneration. Since current treatment is aimed at preventing symptoms such as dementia and disability, the search for causative therapy that would be helpful in preventing and treating post-ischemic neurodegeneration of Alzheimer’s disease proteinopathy is ongoing. Numerous studies have shown that the high contents of flavonoids and phenolic acids in honey have antioxidant, anti-inflammatory, anti-apoptotic, anti-amyloid, anti-tau protein, anticholinesterase, serotonergic, and AMPAK activities, influencing signal transmission and neuroprotective effects. Notably, in many preclinical studies, flavonoids and phenolic acids, the main components of honey, were also effective when administered after ischemia, suggesting their possible use in promoting recovery in stroke patients. This review provides new insight into honey’s potential to prevent brain ischemia as well as to ameliorate damage in advanced post-ischemic brain neurodegeneration.

## 1. Introduction

Ischemic stroke is a disease characterized by high mortality, morbidity, recurrence, and low cure rate [1]. Worldwide, ischemic stroke is the leading cause of severe long-term disability and the second leading cause of death (5.5–6.0 million) after ischemic heart disease [2,3,4,5,6,7,8]. Ischemic stroke, caused by occlusion of an artery leading to the brain, is the most common form of stroke, accounting for roughly 85–90% of all strokes [3,5,8,9,10,11]. The incidence of stroke increases with age in both men and women [7], but in some countries, especially India and China, the incidence of stroke in people under 40 has recently increased, representing a serious problem for public health [12]. In the coming years, the global trend of extending life expectancy will cause a parallel increase in the incidence of stroke.

Approximately 70% of ischemic strokes and 87% of stroke-connected deaths and disability-adjusted life years occur in low- and middle-income countries [13]. In these countries, the number of ischemic stroke cases has more than doubled in the last four decades [13]. In contrast, over the last forty years, the incidence of ischemic stroke decreased by 42% in developed countries [13]. Ischemic stroke occurs on average 15 years earlier and causes more deaths in people in developing countries compared to people in developed countries [14]. As many as 84% cases of ischemic strokes in developing countries die within three years of stroke, compared with 16% in developed countries [13]. Current epidemiological statistics evaluate that approximately 14–17 million people suffer from ischemic stroke annually, of whom approximately half die within a year [2,3,4,5,6,7,8,15]. It is now believed that one in six people in the world will suffer a stroke in their lifetime [15]. It is also known that the number of post-ischemic patients all over the world has reached approximately 33 million [2,6]. Interestingly, in the 21st century, the number of cases of ischemic stroke in young adults has increased to about 2 million per year [16]. Due to aging and obesity among adults in the European community, it is estimated that by 2025, the incidence of stroke in this group will increase to 1.5 million [16].

There has been a global decline in age-adjusted mortality and stroke rates for the last 25 years, but the total number of stroke cases has increased as life expectancy has increased [10]. Ischemic stroke is the most common cause of brain ischemia in humans in developed countries, and the number of cases is 10 times higher than that of hemorrhagic stroke, while in developing countries this difference is much smaller [10]. Although ischemic stroke death rates are dropping, it is believed that up to 50% of stroke-connected deaths can be attributed to improper preventive or medicinal management of modifiable risk factors [10]. The risk of recurrence of ischemic stroke during the first month of recirculation is high; 1 in 25 patients have been shown to have a repeated stroke in this time frame [10]. According to the latest forecasts, the number of ischemic stroke cases will increase to 77 million in 2030 [2,6]. If the tendency of growing ischemic stroke indicators in the world continues, by 2030 there will be about 12 million deaths worldwide, 70 million patients will suffer a stroke, and more than 200 million disability-adjusted life years will be recorded annually [6].

Despite progress in diagnosis and symptomatic treatment of ischemic stroke, it is calculated that the number of strokes will more than double by 2050, and long-term disability following stroke will increase in equal measure due to demographic shift and the growing amount of stroke survivors [17,18]. It is estimated that 30–40% of ischemic strokes cases are cryptogenic, i.e., they have no known cause [9]. Symptoms usually vary depending on the extent of the stroke and the area of the brain affected, and include sensory and motor dysfunctions that are often permanent. Approximately 30–50% of stroke survivors do not return to functional independence [9].

Brain ischemia has been shown to trigger a sequence of phenomena called the “ischemic cascade” that can last from minutes to days [9,19]. These phenomena include energy failure, excitotoxicity, oxidative stress, disruption of the blood–brain barrier, neuroinflammation, and ultimately cell death [8,9,20]. Ischemic lesions cause cortical and subcortical infarcts, white matter damage, cerebral amyloid angiopathy, and microbleeds [21,22,23]. This additionally causes hypoperfusion, ischemia of adjacent structures, chronic neuroinflammation, accumulation of amyloid plaques and neurofibrillary tangles, gliosis, neuronal death, and finally brain atrophy [24,25,26,27,28,29,30]. Post-ischemic neurodegeneration commonly involves injury to the following brain regions: the cerebral cortex, temporal lobe, hippocampus, amygdala, entorhinal cortex, and parahippocampus. After ischemia, these regions are involved in memory and cognitive deficits, and their progressive neurodegeneration also triggers behavioral changes. Since most regions affected by pathology are related to both cognition and behavior, this makes behavioral changes strongly correlate with cognitive dysfunction. Ischemia-related cognitive impairment ranges from mild to severe, occurring in about 35–70% of survivors one-year post-stroke, with higher rates seen soon post-stroke [31,32,33,34,35,36]. Approximately 20% of patients with mild cognitive impairment post-stroke make a full recovery, with the highest rate of recovery seen soon after insult [36,37]. However, cognitive improvement without returning to pre-stroke levels is more common than fully recovery [36,38,39]. Interestingly, the risk of developing dementia in the future increases after ischemic brain damage, even in patients with transient cognitive impairment [36,40]. Brain ischemia has been shown to speed up the beginning of dementia by 10 years [41]. Approximately 8–13% of patients experience dementia soon after a first ischemic stroke and more than 40% after a second ischemic stroke [6,35,41]. In addition, the estimated progress of dementia in patients surviving 25 years post-stroke is approximately 48% [24,41].

It should be emphasized that patients experiencing an ischemic stroke have significant implications for caregivers, society, and the economy, especially in developing countries where the projected increase in the incidence of stroke is the highest. Not surprisingly, the socioeconomic influence of strokes is huge and increasing over time, with an annual cost in the EU of EUR 38 billion in 2012, EUR 45 billion in 2015 and EUR 60 billion in 2017 [42].

In view of the above data, there is an urgent need to develop new therapies capable of preventing or reducing brain ischemia related to Alzheimer’s disease proteinopathy damage. Currently, there are no causal treatments available that could prevent the disease or effectively treat its sequelae. It is understood that ischemic prophylaxis should be started as early as possible to reverse the natural progression of the disease. Currently, of course, reperfusion remains the only and immediate therapeutic option in ischemic stroke. At the present, despite limited effectiveness, thrombolysis with recombinant tissue plasminogen activator (rtPA) and thrombectomy are used in the therapy of acute stroke [5]. Less than 10% of ischemic stroke cases receive rtPA treatment [43]. However, the clinical profit is less than three percent due to the restricted time window for optimal medical therapy and the potential for extensive bleeding following drug administration [44,45]. In addition, reperfusion strategies, despite their limited effectiveness, are applicable only to a small percentage of patients due to the short time window of therapy, contraindications, and costs connected with maintaining the infrastructure for performing procedures [5]. In addition, the possibility of further therapeutic intervention after reperfusion procedures in the form of a neuroprotective effect is now proposed. Neuroprotection is not an alternate to thrombolytic treatment and thrombectomy, but aims to brake alternations in brain parenchyma and preclude the spread of damages to adjacent areas or structures. The idea of neuroprotection is promising in preclinical studies but has not been translated into clinical success [46,47]. A fresh notable example is nerinetide, which interferes with postsynaptic density protein 95, an excitatory neuronal protein. In experimental studies, it has shown promise in the therapy of cerebral ischemia, but has not shown treatment advantages in human stroke [47]. One of the reasons why neuroprotective strategies fail is that they mainly target neuronal cells. It should be added that the brain is composed of different cells, i.e., astrocytes, microglia, oligodendrocytes, endothelial cells, and pericytes, all of which influence the function and survival of neuronal cells by releasing different substances that act either positively or negatively. These cells are under-studied as therapeutic targets in post-ischemic stroke, but understanding of their post-ischemic behavior, both harmful and beneficial, continues to grow and is being intensively studied.

Therefore, in this review, we will focus on the pleiotropic protective effects of honey on persistent neurons after ischemia and on the neuropathological phenomena that develop following ischemic stroke and experimental brain ischemia. On top of this, among all the above-mentioned needs, honey stands out for its high therapeutic potential for almost all aspects of post-ischemic neurodegeneration, such as oxidative stress [48], neuroinflammation [49], neuroprotection [50,51], and cholinergic function [52], as well as memory and cognition [53,54]. This review focuses on natural neuroprotective compounds extracted from natural substance such as honey and summarizes their possible effects in the therapy of ischemic stroke.

## 2. Search and Data Collection Criteria

Searches of the literature were performed using the following databases: Scopus, Web of Science, PubMed, and Google Scholar. Keywords used for the literature search were brain ischemia, ischemic stroke, honey, caffeic acid, chlorogenic acid, ferulic acid, ellagic acid, gallic acid, p-coumaric acid, quercetin, kaempferol, luteolin, myricetin, naringenin/naringin, quercetin with ischemic stroke, and apitherapy combination. Works extracted from the databases had to be relevant and up-to-date. Only the latest research was used. The search included mainly works published between 2000 and 2023. Previous original papers of early scientific discoveries are also cited.

## 3. Requirements for Natural Agents in the Treatment of Post-Ischemic Neurodegeneration

As outlined above, post-ischemic neurodegeneration involves many complex neurochemical and neuropathological processes. The first stage of changes occurs during ischemia, and the second during resumption of circulation. Although a considerable number of natural neuroprotective substances have been evaluated in animal models of focal or global cerebral ischemia, only a small proportion of them have been used in clinical conditions, including nimodipine [55,56,57]. The neurochemistry and neuropathology of post-ischemic neurodegeneration is so diverse that it requires simultaneous treatment of multiple therapeutic targets at different stages of neurodegeneration progression. For example, recirculation is restored immediately after brain ischemia with other pathological phenomena characteristic only of circulatory return. In the early stages, therapies should be multicomponent, focusing mainly on reducing oxidative stress, excitotoxicity, apoptosis, and blood–brain barrier permeability, while in the later stages, treatment is required to prevent neuroinflammation and amyloid and tau protein pathology. Treatment types should affect gene activity and protein levels by stopping or limiting gene programming through ischemia.

## 4. Apitherapy

Apitherapy, from the prefix “api-” (from Apis in Latin meaning “bee”) and the word therapy, describes the use of bee products such as honey, pollen, royal jelly, propolis, and bee venom as healing agents [58]. Honey has been used by people since ancient times, by various ancient civilizations (namely the Assyrians, Romans, Egyptians, Greeks, Chinese, Muslims, and Christians), indicating a millennium-long history of using honey as apitherapy [58]. Honey was formerly used as a source of carbohydrates and as a sweetener, but due to its medicinal properties, it began to be used in folk medicine. For several decades, honey has been at the center of interest of researchers conducting preclinical studies to determine the use of numerous properties of honey against various disease entities [58]. The antiseptic, antimicrobial, and wound-healing effects of honey are well documented, and it has antioxidant, anti-inflammatory, immune-enhancing, and anti-cancer potential [58,59]. The physical and chemical properties of honey, such as high acidity and the presence of hydrogen peroxide, may be responsible for the honey’s bactericidal effect, while simple sugars, vitamins, minerals, and enzymes provide nutritional benefits [60,61,62,63]. Despite the growing popularity of honey as an alternative medicine, honey with components such as flavonoids and phenolic acids has not yet become a significant therapeutic agent [58]. The use of honey in diseases has not found a place in modern medicine, with the exception of a few licensed and medically certified honeys for professional wound care in Europe and Australia [59]. Clinical use of honey is not possible due to the lack of clinical studies, as well as the lack of standardization of the use of alternate medicine methods. Although the medical potential of honey has been positively tested in experimental studies [64], for honey to enter the mainstream of medicine as a modern medicament, it must be evaluated in numerous and professional clinical trials with the required special procedures for clinical trials.

## 5. Honey and Its Medical Properties

Honey mostly contains water and sugar. The high sugar content makes it an alternative to glucose. Honey is a natural product which belongs to the most complex foodstuffs. It may be used as a nutraceutical or medicinal supplement apart from being a food sweetener or whole food. Honey is composed of proteins, lipids, carbohydrates, minerals, vitamins and enzymes, carotenoids, organic acids, and aromatic substances. In the past, including ancient times, its main use was associated with sweetening and keeping good health. At present, the biggest honey producers are China (24% of global production) and the EU (14%), and its global production has reached about 1.2 million tones [65]. The highest honey consumption is recorded in Greece (3.5 kg/person/year), the USA (3 kg), Austria (2.5 kg), and Germany (2 kg). In Poland, honey consumption is considerably lower, reaching around 0.65 kg, with linden, multiflorous, and acacia honey being the most preferred types [65]. The medium multiflorous honey price in production is EUR 6.46/100 kg in the EU, the highest price of EUR 19.25/100 kg being encountered in Ireland and the lowest one in Romania (EUR 2.25/100 kg). The multiflorous honey price in Poland (EUR 5.81/100 kg) is close to the EU level [65]. Apart from the honey types mentioned above, exotic honeys (known as Manuka and Malaysian honeys) have recently gained considerable popularity. Although available in health food stores, their prices of about EUR 60 per 500 g limit their consumption. Other honey types popular in Poland (buckwheat and honeydew), as well as Manuka honeys, contain high amounts of polyphenols and antioxidants, which is related to flavonoids and phenolic acids [65]. Evidently, honey’s antioxidant and healing properties result from the presence of phenolic acids and flavonoids [65]. Moreover, the efficacy of antioxidants (vitamin C., uric acid, glutathione reductase, and β-carotene) has been documented to increase in healthy subjects following honey intake at a dose of 1.2 g/kg dissolved in a glass of water and an overall healthy diet [65].

Although the main ingredients do not differ, the minor ingredients of a given type of honey vary significantly and depend on different geographical location, storage conditions, flower source, and final color [63]. Thus, the composition of polyphenolic compounds differs in various honey types and, consequently, in polyphenolic activity and total antioxidant capacity [62]. Evaluation of phenolic content showed that certain types of honey, such as stingless honey and Tualang honey, have higher phenolic acid and flavonoids content and greater antioxidant capacity and radical scavenging activity, which may indicate greater potential in lowering oxidative stress [61]. The medical properties of honeys are primarily linked with phenolic ingredients, mainly flavonoids and phenolic acids, which prevail in honeys [60,66]. Flavonoids include myricetin, kaempferol, quercetin, luteolin, and naringenin/naringin (Figure 1). In contrast, the phenolic acids in honey include gallic acid, ellagic acid, chlorogenic acid, caffeic acid, p-coumaric acid, and ferulic acid (Figure 1) [67]. Honey has the highest content of total phenolic acids (around 2500 mg gallic acid equivalent/kg) and the highest content of total flavonoids (1400–1800 mg catechin equivalent/kg).

According to research on post-ischemic neurodegeneration, all the listed flavonoids and phenolic acids have antioxidant and neuroprotective properties. Since most of the phenolic ingredients, with the exception of chlorogenic acid, influence the pathology of Alzheimer’s disease by reducing amyloid plaques, and naringenin, quercetin, naringin, ellagic acid, and caffeic acid also reduce tau protein level, when the phenolic components are tested, we believe they will show similar potential in post-ischemic neurodegeneration as the Alzheimer’s disease proteinopathy [54]. In this review, the potential efficacy of flavonoids and phenolic acids in the prevention and/or treatment of post-ischemic neurodegeneration of Alzheimer’s disease proteinopathy will be dealt with (Figure 1). Flavonoids and phenolic acids have also been shown to be effective in treating Alzheimer’s disease and its models [54].

## 6. Post-Ischemic Neurodegenerative Cascade in the Brain

Post-ischemic brain injury is a consequence of changes during ischemia and resumption of circulation. It mainly includes primary and secondary, or acute and chronic, lesions. Mechanisms of primary damage include, but are not limited to, damage to the blood–brain barrier, neurochemical changes, and necrotic neuronal death. Mechanisms leading to secondary damage include oxidative stress, lipid peroxidation, neuroinflammation, brain atrophy including the hippocampus, and genomic and proteomic changes. Cerebral ischemia triggers a number of biochemical reactions at the cellular level. They encompass excitotoxicity, oxidative stress, anaerobic metabolism, cellular energy loss, cytoplasmic calcium overload, mitochondrial dysfunction, free radical production, inflammatory reactions, and amyloid and tau protein pathology [6,27,28,30,55,56,57,68,69,70,71,72,73,74,75,76,77]. These processes generate significant pathology of neurons and neuroglial cells and eventually participate in the aggravation of blood–brain barrier dysfunction [57,71,76,78,79,80,81,82,83,84,85,86,87,88,89,90], development of brain edema [81], and brain atrophy [27,82,83,84], which ultimately results in irreversible brain damage [27] and dementia [84,85,86,87].

Post-ischemic injury is characterized by a genetic, neurochemical, and neuropathological cascade of events in which an intense and prolonged inflammatory response plays a key role in the further progression of brain injury [28,30]. Abundant evidence shows that post-ischemic neuroinflammation is connected with acute and chronic blood–brain barrier disruption, vasogenic edema, hemorrhagic transformation, and poorer neurological outcomes in animals and humans. While inflammation contributes to brain injury in the early stages of ischemic insult, the inflammatory response may promote recovery in the late stages by facilitating neurogenesis, angiogenesis, and neuronal plasticity. A better molecular understanding of how the inflammatory response progresses from injury to repair would help identify new strategies for therapeutic intervention in time- and context-specific ways.

### 6.1. Excitotoxicity

In fact, focal or global brain ischemia is caused by a local or complete cessation of blood supply to the brain, resulting in the transition of the ischemic area on anaerobic metabolism. Anaerobic metabolism leads to a depletion of adenosine triphosphate and an accumulation of lactic acid. This condition is referred to as lack of energy and occurs immediately after cerebral blood flow is stopped. At the cellular level, the lack of energy results in the inability to pump ions outside the neuron, and further, excessive accumulation of sodium and calcium ions with the following influx of water into the neuron [68,81]. As a consequence, cell swelling or cell lysis occur under the influence of excess water [81]. The cell membrane is depolarized, which results in the opening of voltage-sensitive calcium channels, inducing a massive influx of calcium into the neuronal cell [68]. An excess of intracellular calcium ions stimulates the production and release of excitatory amino acids, especially glutamate [68]. Next, elevated glutamate activates N-methyl-D-aspartate receptors [68,69,72]. Activation of these receptors further increases the influx of calcium into cells affected by ischemia. Finally, an increase in intracellular calcium promotes the generation of reactive oxygen species, the release of pro-radicals such as free ions, and the synthesis of excess nitric oxide by neuronal nitric oxide synthase, which is modulated by the N-methyl-D-aspartate receptor [88]. In addition, the activation of proteases, lipases and endonucleases is increased, which causes the release of fatty acids and damage to the cell membrane and triggers a cascading series of reactions leading to cell death by apoptosis [88].

### 6.2. Neurotransmission

Acetylcholine is a neurotransmitter that plays a key role in the transmission of neuronal signals and memory formation. Lack of acetylcholine, especially in the hippocampus, is manifested by post-ischemic dementia [89]. Acetylcholinesterase lowers acetylcholine levels after ischemia in the brain [89]. In addition, the inflammatory factor IL-1 can increase the level of acetylcholinesterase and accelerate the breakdown of acetylcholine, causing a decrease in the level of acetylcholine in the brain, which affects the ability to remember [90].

Acetylcholine, being the main neurotransmitter of cholinergic neurons, has also been shown to play a key role in the development of post-stroke inflammatory changes [91,92]. After ischemia, the cholinergic anti-inflammatory pathway is activated, which blocks the generation of pro-inflammatory cytokines, such as TNF-α [93,94]. Acetylcholinesterase is known to induce apoptosis [95,96] and low acetylcholinesterase levels are responsible for the stimulation of the acetylcholine dependent anti-inflammatory process in post-stroke patients [94]. Acetylcholinesterase activity peaks on the second day after local brain ischemia, while the return to control values is noted on the eighth day of recirculation [95]. Activation of the acetylcholine receptor is known to inhibit the generation of TNF-α, IL-1, and pro-inflammatory cytokines that induce the inflammatory effect of microglia [45,97]. Acetylcholinesterase activation has been shown to occur earlier than microglial activation [95,96]; despite this, a direct relationship between them has been found [98].

Honey improves the functions of the cholinergic and glutamatergic systems [52,54,99]. As already underlined, amyloid is known to impair the functioning of glutamatergic neurons due to a massive influx of calcium ions into the cell, which results in overstimulation and release of acetylcholinesterase, causing a reduction in choline acetyltransferase and acetylcholine [89]. This is confirmed by results obtained in the Alzheimer’s disease model, where honey administration has significantly diminished inflammation and oxidative stress, reducing the activity of acetylcholinesterase as well [54]. The available evidences indicate that the synthesis of choline acetyltransferase and acetylcholinesterase is diminished with age, this effect being positively correlated with the development of dementia [100]. Further, honey augments the transport of choline, significantly affecting the synthesis of acetylcholine [54]. Indeed, honey administration has been documented to decrease the glutamate-mediated excitatory response in the striatum by apparent reduction of calcium influx via voltage-operated and NMDA-gated calcium channels, leading to anti-excitotoxicity effects [54]. It should be noted that gamma-aminobutyric acid (GABA) has an inhibitory effect on the mature nervous system, affecting the normalization of neurotransmission through interaction with glutamate [101]. Enhanced GABA signaling inhibits ischemia-induced glutamate release affecting the normalization of these neurotransmitter pathways [101]. Post-ischemic imbalance, i.e., reduced GABA signal transmission and increased release of excitatory glutamate, indicates that GABA plays a special role in post-ischemic recovery of neurotransmission [68,101]. Increasing the level, and thus the activity, of GABA, e.g., by using honey or its ingredients, may partially restore the balance of GABA and glutamate transmission after brain ischemia, indicating a potential strategy for the prevention and/or treatment of post-ischemic pathological neurotransmission [101]. The effect of honey to increase GABA levels simultaneously inhibits ischemia-induced glutamate release [68], so honey treatment can independently simultaneously stimulate GABA and inhibit glutamate release, resulting in an enhanced dual action on both neurotransmitters, ultimately leading to the normalization of both neurotransmission pathways [101].

### 6.3. Blood–Brain Barrier

The blood–brain barrier is a highly selective structural and functional barrier between the blood and the brain, creating a unique microenvironment required for the proper functioning of the brain and maintaining homeostasis. Endothelial cells of the blood–brain barrier interact with astrocytes, pericytes, platelets, microglia, and perivascular macrophages in the neurovascular network [78,79,81,102]. The concept of a neurovascular network has been proposed to highlight the complex interactions between all cells associated with the blood–brain barrier. An ischemia-reperfusion episode is responsible for a series of events augmenting the permeability of the blood–brain barrier to cellular and acellular blood components, and leading to the opening of tight junctions. Sometimes diffuse leakage of all blood elements via the necrotic vessel wall may also occur [78,103,104,105,106,107,108]. After ischemia, two characteristic features in the altered blood–brain barrier deserve attention. One feature is important due to the chronic activity of extravasated substances [106,107], such as neurotoxic amyloid and tau protein, in producing irreversible neurodegeneration. The other feature is related to the leakage of blood cellular elements, such as platelets, which are involved in acute and massive destruction of the brain tissue [75,79,108,109]. Because amyloid and tau proteins are able to cross the damaged blood–brain barrier, they may produce local neurotoxic effects to ischemic neurons, including increased amyloid production and deposition in the brain in a vicious circle [110,111,112]. After an ischemic episode, blood-derived amyloid can be delivered to ischemic brain tissue, and thus may cause the development of cerebral amyloidosis and cerebral amyloid angiopathy [21,22,26,108,110,111,112,113,114]. Available data indicate that one of the most important neuropathological changes following the development of brain ischemia is a significant increase in the permeability of the blood–brain barrier, ultimately leading to irreversible consequences in clinical conditions [41,76,115].

### 6.4. Neuroinflammation

Neuroinflammation plays a key role in neuroprotection and recovery after brain ischemia [76,116]. Post-ischemic neuroinflammation is triggered by free radicals and dying neuros and is accompanied by cytokine production, activation of astroglial and microglial cells, and infiltration of peripheral immune cells into the brain, which in later stages culminates in the development of a glial scar surrounding the damaged area [6,117,118]. This phenomenon activates positive pro-inflammatory feedback, which intensifies ischemic damage to the parenchyma [6,119]. The neuroinflammatory response initiated by experimental brain ischemia lasts up to two years [28,30] and interacts with neurogenesis and post-ischemic neuronal repair [120,121]. In this regard, the interaction between neuroinflammation and the regenerative response has been extensively studied, and whether this is a beneficial or harmful reaction is still unexplained [6]. The important role of the neuroinflammatory process in the acute and chronic phases of post-ischemic injury has led to the search for factors modulating this pro-inflammatory loop as a putative palliative and regenerative strategy [116].

Ischemia-damaged neurons and neuroglial cells and activated endothelial cells produce cytokines—interleukins and tumor necrosis factor-α—which cause a neuroinflammatory response resulting in secondary energy failure and apoptosis-dependent delayed neuronal death [27,122]. Secondary energy failure may be usually found between 6 and 48 h post-ischemia, leading to the development of necrosis and apoptosis which can last for days or even weeks following the ischemic insult [122,123]. Eventually, the progression of ischemic damage enters the tertiary period of neuropathology by enhancing neuroinflammation and reducing neurogenesis, synaptogenesis, and axonal growth [124,125].

### 6.5. Free Radicals

Free radicals, encompassing reactive nitrogen species and reactive oxygen species, are involved in post-ischemic neurodegeneration [88,126]. The concept of “oxidative stress” assumes an imbalance between the production of oxidants and antioxidants, which can induce damage to the brain and peripheral tissues [88]. Resumption of circulation causes additional post-ischemic damage to the brain by reactive oxygen and nitrogen species, resulting in damage to mitochondria, membrane, and organelles [88,126]. Dysfunctional mitochondria produce increased amounts of reactive nitrogen and oxygen species in a vicious circle. Further, hypoxanthine and pro-radicals, such as iron, are associated with the production of reactive oxygen species [88]. The most significant and last phenomenon involved in apoptotic neuronal death is mitochondrial destruction. The phase consisting of the above phenomena is known as reperfusion injury and takes place within hours of resumption of circulation.

### 6.6. Amyloid and Tau Proteins

Increased levels of staining of various parts of the amyloid protein precursor [27,82,127], lowered levels of non-amyloidogenic α-secretase [74], and accumulation of amyloid in the form of amyloid plaques [25,26] with activation of microglia and astrocytes was found in the brain during survival up to 2 years after ischemia [28,30]. This directly indicates that the neuropathological changes, including changes in the tau protein, triggered by brain ischemia are identical to those in Alzheimer’s disease [29,41,107,115,128,129,130]. In our previous studies, we assessed the expression of genes related to amyloid generation in the CA1 and CA3 areas of the hippocampus and temporal cortex [77,88]. Alzheimer’s disease-associated genes such as *amyloid protein precursor*, *β-secretase*, and *presenilin 1* and *2* undergo ischemic genetic programming. The expression pattern in the CA1 area of the hippocampus covers all genes tested, and the changes persist for the duration of survival (30 days) [131]. In contrast, the pattern of gene expression in the CA3 region was much less severe, did not always occur after ischemia, and was delayed in time after ischemia compared to the CA1 area [74]. Conversely, a pattern of significant gene expression in the temporal cortex appeared immediately after ischemia but did not occur at all survival times and did not affect all genes [132,133]. After reversible brain ischemia, evidence suggests that various forms of dysregulation of the *amyloid protein precursor*, *β-secretase*, and *presenilin 1* and *2* genes are related with different individual neuronal responses in CA1 and CA3 areas of the hippocampus and temporal cortex. Evidence has shown that an ischemic episode of the brain launches amyloidogenic processes. In addition, an ischemic episode further activates amyloid-mediated neuronal death in CA1 and CA3 areas of the hippocampus and temporal cortex. Also, clinical studies have shown an increase in the level of amyloid in the blood in patients with a history of brain ischemia [113,114,134]. Increased plasma amyloid levels in these patients were found to be detrimentally correlated with post-ischemic neurological outcomes [134]. This evidence indicates that post-ischemic accumulation of amyloid is responsible for the progression of neurodegeneration through chronic and massive neuronal death [75].

Elevated tau protein level assessed by microdialysis during cerebral ischemia [135] correlated very well with tau protein levels in the blood and cerebrospinal fluid of post-ischemic patients [136,137,138,139,140,141,142]. In addition, hyperphosphorylated tau protein in the form of neurofibrillary tangles is present in the post-ischemic cytoplasm of neurons, and this phenomenon is the terminal element of neuronal death after ischemia [29,128]. Glycogen synthase kinase-3 involved in tau protein hyperphosphorylation has been found to be increased after ischemia [129]. Hyperphosphorylation of tau protein mediated by glycogen synthase kinase-3 is attributed to the presence of amyloid, which inhibits insulin receptors, finally triggering activity of glycogen synthase kinase-3 [143]. The aggregates that result from the self-assembly of the tau protein further complicate the state of neurodegeneration by destroying neuronal and neuroglia cells [144]. The presence of hyperphosphorylated tau protein and its aggregates in neuronal cells induce cell death via apoptosis [143]. In addition, there are evidences that hyperphosphorylated tau protein binds to normal microtubule-associated protein tau, resulting in synaptic degeneration [145].

Following ischemia, tau protein loses its propensity to form microtubules and begins to self-assemble, inducing the development of paired helical filaments [130], straight fibers, or twisted ribbons [129] that together evolve into neurofibrillary tangles [29,128,146]. Along with the association of neurofibrillary tangles with synaptic dysfunction and neuronal death, neurofibrillary tangles density strongly correlates with cognitive decline after ischemia [29,128,146]. Post-ischemic tau protein also undergoes further post-translational alterations, such as ubiquitination, acetylation, glycation, and cleavage [77,147]. It should be mentioned that aggregated tau protein is more difficult to dephosphorylate by protein phosphatase 2A. Tau protein, like prion disease, has been shown to directly infect adjacent and connected by synapses neurons, and is also released into the extracellular space in its naked form [148,149,150]. Once released, it interacts with low-density lipoprotein receptor-bound protein, heparan sulfate proteoglycans, or muscarinic receptors and is internalized by other neurons through endocytosis and tunneling of nanotubes connecting the cytoplasmic contents of neighboring neuronal cells [151]. After inoculation, it induces the aggregation of natively folded tau protein in naive cells, triggering cellular toxicity and the spread of prion-like pathology [148,149].

### 6.7. Vasospasm

Transverse and longitudinal cerebral blood vessel vasoconstriction after transient cerebral ischemia have been documented [102,105,152]. Arteries and arterioles narrow in circumference, and veins and venules narrow shortening in length [105]. In addition, cerebral vasoconstriction is caused by amyloid, which appears after ischemia in excess in brain tissue and blood [153]. Vasoconstriction is characterized by massive folding of the endothelium and basement membrane, narrowing of the pericytes, growth of microvilli on the endothelial surface, and the involvement of aggregating platelets [79,102,105,154,155]. As an effect, aggregating platelets create microthrombi fastened to the destroyed endothelium, leading to a constant inflow of constrictive matters like thromboxane A2 or amyloid [70,71,153,155]. Due to the fact that amyloid triggers vasoconstriction in vivo, it is believed that it plays a key role in the development of cerebral hypoperfusion post-ischemia. These data, at least in part, explain the crucial role of post-ischemic cerebral amyloid angiopathy, operating in a vicious circle, in chronic and progressive reduction of cerebral blood flow until its complete cessation [21,22]. Thus, amyloid leads to permanent vasoconstriction by accumulating in the vascular walls and reducing the production of nitric oxide by the endothelium, consequently leading to hypoperfusion or lack of blood flow in these vessels [153]. It is very likely that vasoconstriction related to amyloid vascular accumulation and its properties are key factors in recurrent and repeated ischemic episodes after a first ischemic incident.

### 6.8. Cerebral Amyloid Angiopathy

Cerebral amyloid angiopathy is a pathologic condition associated with the collection of amyloid in the walls of all types of brain vessels. Translocation and deposition of amyloid in the vessel wall deprive vessels of their physiological properties, which is accompanied by other pathologies, such as microbleeding and increased permeability of the blood–brain barrier to pathological molecules [108]. Thus, random and durable post-ischemic insufficiency of the blood–brain barrier may start a chronic processing of accumulation of plasma amyloid, inter alia, in the wall of cerebral vessels [110,111,112,113,114,134]. This phenomenon has been termed cerebral amyloid angiopathy and is characteristic of post-ischemic neurodegeneration [21,22] as well as Alzheimer’s disease [54].

Among other things, the accumulation of collagen and the thickening of the basement membrane in the vessels after cerebral ischemia is related to the deposition of amyloid in the vascular wall [156]. This process affecting the vascular wall after ischemia can result in degeneration of the endothelium and pericytes, which further contributes to chronic failure of the blood–brain barrier. In turn, dysfunction of the blood–brain barrier leads to the accumulation of further amounts of amyloid from the plasma, causing irreversible amyloid deposition and vascular degeneration. Under these conditions, amyloid from the blood also interacts with all parts of the vessel wall. It is worth stressing that the altered brain large vascular network may become a seed for amyloid plaques. It is very probable that amyloid plaques are related to vessels exposed to damage induced by cerebral ischemia. Massively accumulated amyloid seems associated with atrophy and rupture of capillaries, which results in free amyloid cores in the brain parenchyma [25]. In this context, the core of senile amyloid plaques is a consequence of the predominant amount of amyloid which may probably enter the brain from the blood or can result from vascular atrophy [25,110,112]. Indeed, capillaries being susceptible to rupture are loaded with amyloid in their walls, which supports the above processes. Eventually, cores with excess amyloid transform into senile amyloid plaques [25].

Following ischemia in experimental and clinical conditions with development of cerebral amyloid angiopathy, microbleeding in the brain parenchyma has been found [23]. Hemorrhage recruits and activates platelets at the rupture site, leading to a vicious cycle. At this point, it should be particularly accentuated that there is huge amount of the amyloid protein precursor, as well as amyloid itself, in the platelets, which plays a key role in the induction of cerebral amyloid angiopathy [108]. One can conclude that the accumulation of blood- and platelet-derived amyloid in the vascular wall shortly after ischemia is a factor inducing cerebral amyloid angiopathy. This phenomenon may be treated as an irreversible vicious circle following ischemia-produced brain injury.

### 6.9. Neuronal Death

The death of brain neurons occurs immediately after ischemia in the form of necrosis, while in later survival times as delayed neuronal death (i.e., apoptosis) [27,122]. Apoptosis or programmed neuronal death is an important phenomenon of delayed ischemic neuronal death. Necrotic neuronal death which occurs immediately post-injury is the predominant death following ischemia [27,122]. It has been found that the phenomenon of apoptosis combined with necrosis of neuronal cells can last even years after the initial ischemic damage [27]. Cerebral ischemia in animals and humans mainly causes irreversible damage to the pyramidal neurons of the hippocampus, which is responsible for memory, and the cerebral cortex especially in temporal lobe, which finally results in generalized brain atrophy like in Alzheimer’s disease [27,54,82,83,84].

### 6.10. Dementia

Currently, over 30 million patients suffer from the consequences of brain ischemia [157]. At the present time, brain ischemia in humans is a major medical and public challenge, as it is strongly associated with age-related mild cognitive impairment. Some research has shown stable rates of cognitive impairment in the time interval 3 months (22%) to 14 years (21%) following ischemia [32]. It has been shown that the risk of dementia in patients with a history of ischemic stroke is twice as high [1,87]. Current data indicate that following an ischemic stroke, cognitive impairment and dementia may touch up to one-third of stroke survivors [1,87]. The percentage of patients with full-blown dementia was shown to increase by 157% following stroke and will proceed to grow with recurrence and severity or volume of ischemic episode [86]. Despite the decline in the incidence of stroke over time, dementia remains a serious problem in post-stroke patients [1,86]. Both ischemic stroke severity and its recurrence are connected with an increased risk of dementia, highlighting the importance of not only stroke prevention but also secondary post-stroke prevention to decrease stroke recurrence. According to epidemiological data, these evidences do not show that treating or preventing ischemic stroke would restrict the risk of dementia, but they do support the value of interventions in reducing stroke size and its severity [1,86]. Future research should clarify whether healing activities to reduce stroke severity and improve secondary prevention have an effect on dementia incidence, and should control patients for a longer period after stroke to assess the ultimate impact [1]. Intravenous thrombolysis after a first ischemic stroke was found to reduce the incidence of dementia by 24% [1].

Overall, this review highlights the multifactorial and multiphenomenal nature of brain ischemia, as well as the complexity and multidirectionality of its pathways, emphasizing the need to simultaneously capture and eliminate multiple risk factors and pathological mechanisms at an early stage of disease, and pointing to the urgent need for new diagnostic strategies to capture important early risk factors, as well as for more accurate and early diagnosis. The lack of ultimate effectiveness of currently used drugs and therapeutic methods, as well as the complex multifactorial pathogenesis before, during, and after ischemia pose new challenges to the scientific community regarding the development of drugs, based primarily on the paradigm of creating a multidirectional medicine [158,159].

## 7. Therapeutic Potential of Honey and Its Ingredients in Post-Ischemic Neurodegeneration

Honey administered orally before and after brain ischemia reduced damaged pyramidal neurons in the hippocampus, and significantly improved spatial learning and memory performance [50,160].

Quercetin and its glycosides, including isoquercetin, have a beneficial effect on pathological changes in various models of ischemic brain injury [161,162,163,164,165,166,167,168,169]. The beneficial effect of quercetin results from its anti-inflammatory, anti-apoptotic, and antioxidant effects, and from inhibiting metalloproteinase activity, which prevents blood–brain barrier failure after cerebral ischemia [161,162,163,164,165,166,167,168,169]. It is suggested that the factor Nrf2 may be involved in the antioxidant and anti-apoptotic effects [161,162,163,164,165,166,167,168,169].

Myricetin has been studied for multiple medical effects, including anti-apoptotic, anti-inflammatory, and antioxidant properties [170,171]. Studies have shown that myricetin works against brain damage after local ischemia [170,171,172]. Among the proposed molecular mechanisms of myricetin action, inhibition of p38 MAPK and enhancement of AKT and Nrf2 factors have been indicated [171,173].

A protective effect has been demonstrated with kaempferol in transient local cerebral ischemia in rats, e.g., in reducing amyloid protein precursor [174,175,176]. Studies indicate that treatment with kaempferol after ischemia prevents the development of neuroinflammation by reducing the activation of NF-kB/RelA and STAT3 [174,175,176].

Naringenin exhibits neuroprotective properties in a model of cerebral ischemia, reducing apoptosis, inflammation, oxidative stress, and neurological deficits by modulating claudin-5, MMP9, and Nrf2 [177,178,179]. In addition, naringin, a naringenin-7-O-glycoside, prevented brain microvascular thrombosis in spontaneously hypertensive rats [64].

Luteolin has been shown to have a neuroprotective effect, i.e., anti-apoptotic and blood–brain barrier stabilization, on ischemic brain damage by increasing claudin-5 and inhibiting MMP9, reducing oxidative stress, and enhancing autophagy by activating the Nrf2 pathway and reducing inflammatory changes [180,181,182,183].

Administration of caffeic acid before or after ischemia had a protective effect on the brain by improving the neurological outcome in various models of cerebral ischemia [184,185,186,187,188]. The neuroprotection provided by this substance was probably mediated by the inhibition of 5-lipoxygenase and both antioxidant and anti-inflammatory effects [64,184,185,186,187,188].

The beneficial effect of ferulic acid was confirmed in animal models of global and local brain ischemia [189,190,191,192,193,194]. There are many indications that the neuroprotective effect of ferulic acid is related to the anti-inflammatory and neurotrophic effects related to a reduction in the activity of intercellular adhesion molecule-1, an increased level of erythropoietin in the brain, and granulocyte colony-stimulating factor [64,189,190,191,192,193,194]. In addition, it has been shown that ferulic acid extends the therapeutic window after focal cerebral ischemia, which is currently very useful in the clinic [191].

On the other hand, p-coumaric acid has shown neuroprotective effects in models of local and global cerebral ischemia by inhibiting apoptosis and the production of reactive oxygen species [195,196].

However, chlorogenic acid given before or after ischemia diminished infarct size, blood–brain barrier damage, and behavioral deficits in focal cerebral ischemia by affecting the activation of MMP, increasing the levels of erythropoietin, HIF-1α, and nerve growth factor in the brain [197,198,199,200,201,202,203]. In addition, the substance supported neuroprotection in rats by affecting the Nrf2 path in a model of brain ischemia induced by ligation of both common carotid arteries [199]. It is worth noting that the other substance contained in honey, chlorogenic acid in combination with rtPA, effectively reduced behavioral deficits in focal cerebral ischemia in rabbits and extended the time window of rtPA treatment [64,204].

Another honey ingredient, ellagic acid, had a protective effect after experimental ischemic brain injury by affecting the regulation of Bcl-2/Bax activity [205].

Gallic acid protects against experimental, transient focal, and global ischemic brain injury by decreasing oxidative stress with elevated antioxidant levels and reducing markers responsible for inflammatory answer [206,207,208,209,210]. The neuroprotective effect of gallic acid is attributed to its ability to enter the brain through the blood–brain barrier, directly decreasing the concentration of reactive oxygen and nitrogen species and chelating transition metal ions [210]. Gallic acid has been documented to interrupt the vicious cycle of oxidative stress during brain injury due to ischemia [206,207,208,209,210].

In addition to the primary anti-inflammatory, antioxidant, and anti-apoptotic potential shown above, most honey ingredients, such as luteolin, myricetin, naringenin, quercetin, kaempferol, caffeic acid, ellagic acid, ferulic acid, gallic acid, and p-coumaric acid, also have a therapeutic effect on the progression of neurodegeneration associated with amyloid pathology in Alzheimer’s disease [54] and ischemia-related brain neurodegeneration of Alzheimer’s disease proteinopathy [175,176,177]. Additionally, naringenin, quercetin, naringin, ellagic acid, and caffeic acid also decrease tau protein phosphorylation in the brain in models of Alzheimer’s disease [54]. Honey components reduce the expression of genes involved in amyloidogenic neurodegeneration, such as *amyloid protein precursor*, *β-secretase*, and *presenilin 1* [54,211,212,213]. In addition, myricetin and ellagic acid prevent the production of amyloid by increasing the expression and activity of α-secretase, which leads to a decrease in the cleavage of the amyloid protein precursor to soluble amyloid, and thus prevents the synaptic deposition of the latter [211,212]. These data suggest that the phenolic ingredients of honey participate in the regulation of the expression of genes involved in the reduction of oxidative stress and the development of amyloid fibrils [54]. Also, flavonoids and phenolic acids elevate the expression of the Nrf2 transcription factor, which is in charge for the induction of antioxidant genes, thus improving protection against oxidative damage [54]. Additionally, by reducing the induction of inflammatory factors, honey ingredients also weaken the immune response of microglia and astrocytes in the hippocampus, entorhinal cortex, and amygdala. Further, honey ingredients reduce tau protein hyperphosphorylation, which prevents the development of neurofibrillary tangles [214] and reduces the production and accumulation of amyloid in the form of plaques [215,216]. They also appear to exert neuroprotective effects by preventing neuronal damage and apoptosis and regulating the cholinergic system, similar to curcumin analogs in scopolamine-induced amnesia, where they increase acetylcholine and choline acetyltransferase and decrease butyrylcholinesterase [213,217,218]. Chlorogenic acid, ellagic acid, caffeic acid, gallic acid, myricetin, naringenin, quercetin, and kaempferol lower the level of acetylcholinesterase [54] like curcumin analogs in an experimental model of amnesia [217,218]. Additionally, caffeic acid and gallic acid also reduce level of butyrylcholinesterase [54]. It has also been shown that the action of chlorogenic acid leads to a reduction in memory and cognitive deficits in humans [219]. Since honey contains many flavonoids and phenolic acids, it can be expected that its consumption and proper preparation as a medicinal substance will have great potential in the prevention and/or treatment of post-ischemic brain pathology with the Alzheimer’s disease phenotype. The therapeutic potential of honey and its ingredients in preclinical models of focal and global brain ischemia (regardless of whether the administration was started before, during, or after ischemia), its effective concentration, dose, and duration of treatment, and the main effects are presented in Table 1.

## 8. Bioavailability, Safety and Side Effects of Honey

Bioavailability may be explained as the absorption rate of a substance and its availability to be used inside the organism. The quercetin component of honey, being unstable and poorly absorbed in the digestive tract, may serve as a good example for this term [220]. Quercetin conjugates absorbed in the small amount via the gastrointestinal epithelium undergo rapid liver metabolism, which is responsible for its short biological half-life of 1–2 h [220]. Further, the small unmetabolized quantity of quercetin is available to the organism. Due to their occurring naturally in nature, quercetin glycosides are its most consumed forms. Quercetin does not normally cross the blood–brain barrier [221], but there is a potential for it to enter the brain tissue after ischemia due to chronic opening of the blood–brain barrier [106,107,108]. This phenomenon is most likely to enhance quercetin efficiency in a post-ischemic brain. Thus, increasing its half-life and low bioavailability will significantly favor quercetin’s in vivo effects to attenuate inter alia the excitotoxicity of glutamate [220]. This currently limits its ability to be used in a clinical setting. Therefore, future research should focus on optimizing the conformation of quercetin or developing a quercetin nano-delivery system to improve its bioavailability and rate of penetration through the blood–brain barrier.

Honey has only one serious side effect: a mild hypersensitivity to anaphylactic shock, which may be attributed to the presence of pollen and bee-derived proteins [222]. So far, allergic reactions are rare, as only single cases have been reported [222,223,224]. Honey can be a subject to microbial and extra-microbial contamination [63]. Trace amounts of environmental herbicides, pesticides, heavy metals, and antibiotics given by beekeepers may be also found [63,225]. Additionally, honey can contain poisonous substances, such as the grayanotoxins found in Andromeda Flower Crazy Honey [63]. Several authors have reported that mad honey poisoning presents with asystole, bradycardia, nodal rhythm, and acute myocardial infarction [226,227,228,229]. Thus, the safety of honey production must be ensured in order to meet certain requirements.

Polyphenols have gained the interest of the food industry due to their obvious health benefits, so many phenolic acids and flavonoids have appeared on the market in the form of dietary supplements. Phenolic acids undergo easy absorption in the digestive tract [230,231] and some of them may be available as food products, whereas the main limitation of use of flavonoids is their low bioavailability [232]. Despite this, clinical studies have shown that polyphenol supplements are well tolerated and safe [233].

Future basic studies should aim at identifying and evaluating methods to increase the potential clinical efficacy of, among others, quercetin and honey itself. We believe that the broad analysis of honey and its constituent flavonoids and phenolic acids presented in this review, together with specific approaches to experimental treatment of ischemia-induced neurodegeneration existing in the form of Alzheimer’s disease proteinopathy, may be helpful in further successful human studies in this field.

## 9. Conclusions

Evidence shows that the therapeutic window for a standard effective dose of rtPA can be extended by administering chlorogenic acid, suggesting that it may be most useful at present as a combination therapy with a standard thrombolytic protocol [204]. Next, the data suggest that ferulic acid may provide a therapeutic extension of the time window in ischemia-induced apoptosis, and therefore is a promising substance worthy of further research [191]. In addition, quercetin has been shown to inhibit platelet activation and reduce inflammatory thrombosis in both animal and clinical trials, which is important due to the high content of amyloid protein precursor and amyloid in platelets. This means that the amyloidogenic activity of platelets can be inhibited. This property is important in late post-ischemic survival, where massive accumulation of amyloid and platelets outside the cerebral blood vessels has been demonstrated [79,127]. These results proved that chlorogenic and ferulic acids as well as quercetin have a healing effect on changes after experimental cerebral ischemia and provided a potential theoretical basis for the development of research on these substances and/or honey in clinical trials [201]. Thus, honey ingredients can be used in functional foods or drugs to help treat neuronal dysfunction in ischemia-induced neurodegeneration of the Alzheimer’s disease proteinopathy. Undoubtedly, some potential therapeutic mechanisms of polyphenols in the treatment of ischemic stroke are still waiting to be discovered.

The effectiveness of honey in various in vivo test protocols indicates its pleiotropic therapeutic properties, but before establishing a strategy for using this bee product in the clinic, the following should be investigated: (1) The chemical nature of the honey sample. (2) The effectiveness of honey should be compared with well-known substances, including positive or negative controls in experimental studies. (3) Interactions between honey and other drugs should also be investigated in the clinic. Clinical trials are required to assess the potential of honey in patients and healthy individuals to understand under what conditions honey can support health. The data show the significance of this research area not only for patients and investigators, but also for the pharmaceutical industry, which is looking for new drugs.

Natural substances such as honey show promising neuroprotective properties in the acute phase of experimental cerebral ischemia by inhibiting oxidative stress, reducing neuroinflammation, and suppressing apoptosis. Honey inhibits inflammatory thrombosis, reduces cerebral edema, infarct size, and apoptosis, promotes autophagy, and can be used as an adjuvant in the treatment of ischemic stroke. Although honey has neuroprotective effects in preclinical studies, it has not been successfully translated into clinical settings. One of the main reasons for this failure is that these studies were conducted in healthy young, adult rodents and not in older animals, whereas stroke occurs predominantly in the aging population and older rodents may react differently to natural medicines compared to the neuroprotective effects in young animals. Therefore, it is highly recommended to use older animals to test the effect of natural compounds on changes after ischemic stroke. The results presented in the current paper provide useful evidence to draw attention to honey as a promising natural treatment for post-stroke patients, but the clinical effects need to be proven in the future.

Honey has a protective effect in experimental cerebrovascular diseases, and this effect depends on many factors, with the chemical composition, especially the constituent phenolic compounds, being one of them. The antioxidant properties of honey are probably associated with the synergy among different honey phenolic compounds. Such a relationship may take place regarding other medicinal properties of honey; for instance, inhibition of platelet activity not only in terms of the formation of blood clots, but in the release of amyloid, as well. These effects may be of particular importance in the prevention and treatment of long-term stroke consequences. Based upon the literature, discussing whether honey can be effectively applied in treating cerebral ischemia in humans, some limitations emerge. First, this review takes into consideration only papers dealing with in vitro and animal models. Also, only a small proportion of papers provide data on the chemical properties of honey, especially on the profile of flavonoids and phenolic acid compounds and their effects on ischemia-induced brain neurodegeneration. It is thus difficult to unequivocally assume the significance of honey along with its flavonoids and phenolic acids following ischemia. An additional question arises of whether these substances may possess potential substantial undesired effects, such as bleeding or recurrent ischemia. To conclude, although pure phenolic compounds have been documented to exert antioxidant and often antiplatelet activities, all other potential mechanisms mediated by honey constituents must be fully identified before the true value of honey may be considered as a potential prophylactic and/or therapeutic agent for the management of ischemia-induced neurodegeneration in the form of the Alzheimer’s disease, especially in humans. Despite all of this, it is clear that honey has a major impact on potential post-ischemia outcomes.

## 10. Clinical Perspective

Currently, thrombolysis and thrombectomy are the methods of choice in the treatment of ischemic stroke in humans. Both methods can be called mechanical; they are aimed at removing a mechanical obstacle, i.e., a clot closing a specific blood vessel, and this action, especially in the period after removing the obstacle, is absolutely insufficient and requires pharmacological support. In the absence of causal treatment of stroke, an advancement in the procedure used would be to extend the time window of the thrombolysis and thrombectomy protocol, which would allow more patients to benefit from the current treatment method. Looking at the data, it seems that honey and its components create such an opportunity (Figure 2). Clinical observations have shown a great variety of data obtained from patients with comparable vessel occlusions and cured within the same time window. The results indicate that available treatment methods seem insufficient, and a great number of pathological processes that lead to the progression of brain damage after ischemia remain untreated during the care of stroke patients. We propose a therapy based on honey neuroprotection, which would complement treatment through recanalization. The results of honey’s neuroprotective treatment can be bidirectional. First, honey neuroprotection, applied during and/or immediately after recanalization, can dramatically reduce the volume of primary brain parenchymal damage and rescue the brain from the damage spreads, limit the severity of neuroinflammation, oxidative stress, and glutamate toxicity, and especially mitigate or limit reperfusion injury caused by recanalization. Secondly, the neuroprotective effect of honey may enhance the endogenous phenomena of neuroplasticity and neurogenesis, which will significantly support recovery and reduce the incidence of post-ischemia complications. Evidently, because neurodegeneration expands not only acutely post-ischemia but also progresses within a long period [27], honey, along with its pleiotropic profile, appears a promising adjuvant in the future against ischemia-related development of brain neurodegeneration in the form of Alzheimer’s disease proteinopathy (Figure 2). This review provides a new perspective for clinical trials to consider honey adjuvant therapy to prevent the progression of dementia among stroke survivors.

## Figures and Tables

**Figure 1 molecules-28-05624-f001:**
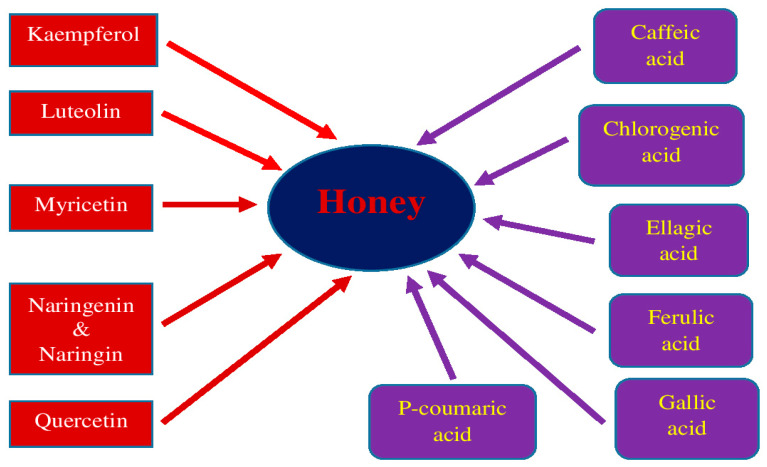
Components of honey, consisting of flavonoids (**left**) and phenolic acids (**right**).

**Figure 2 molecules-28-05624-f002:**
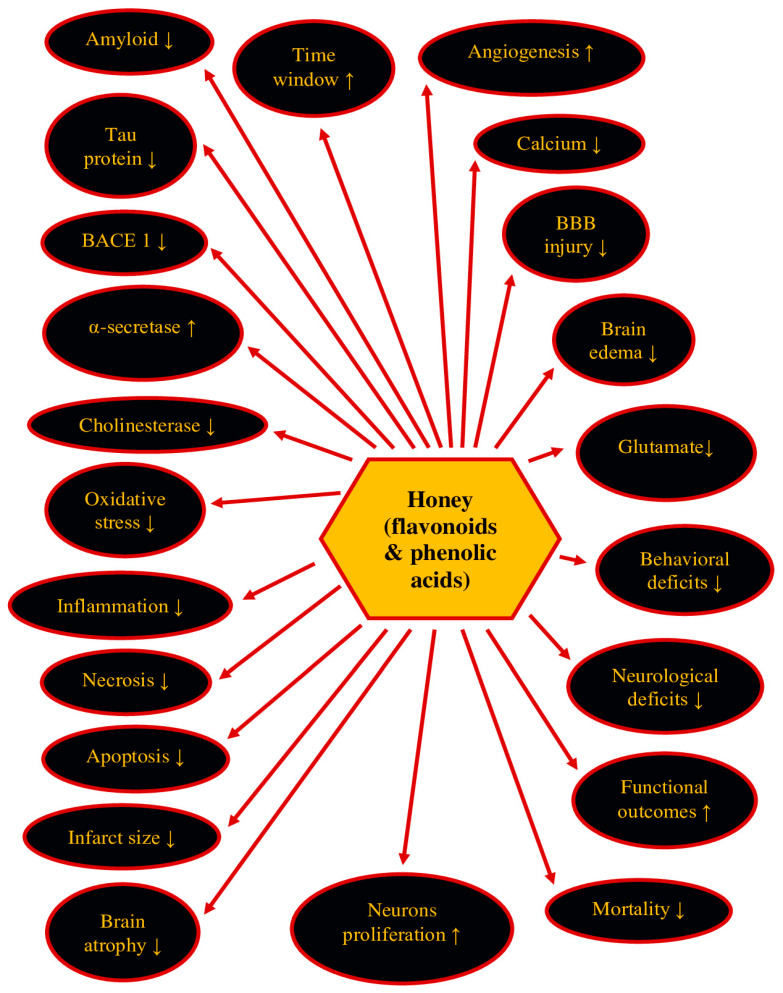
Pleiotropic protective effect of honey on post-ischemic brain neurodegeneration. BACE 1—β-secretase, BBB—blood–brain barrier, ↓—decreased, ↑—increased.

**Table 1 molecules-28-05624-t001:** Activity of honey and selected its flavonoids and phenolic acids in various models of brain ischemia.

Substance	Model	Treatment	Effects	References
		Honey		
Malaysian Tualang honey	p2VO	Pre: 1.2 g/kg for 10 days with Post: 10 weeks	↓ Hippocampal CA1 region damage ↑ Spatial learning, memory performance	[50,161]
		Flavonoids		
Quercetin	tMCAO	Post: 20 mg/kg/d for 3 days	↓ Oxidative stress, necrosis, apoptosis, brain edema, brain injury, neurological deficits	[165]
Quercetin	pMCAO	Post: 30 mg/kg single dose	↓ Brain injury	[162]
Quercetin	Photothrombotic model	Post: 25 µmol/kg every 12 h for 3 days	↓ BBB injury, brain edema, neurological deficits ↑ Functional outcomes	[164]
Quercetin	2VO	Pre: 50 mg/kg 30 min before and immediately post-ischemia, then daily for 2 days	↓ BBB injury, delayed neuronal damage in CA1, CA2, brain injury	[163]
Quercetin	tMCAO	Pre: 10 mg/kg 30 min before	↓ Neurological deficits, behavioral changes ↑ Parvalbumin expression	[169]
Quercetin	tMCAO	Pre: 10 mg/kg 1 h before	↓ Brain edema, damage in brain cortex, neurological deficits ↑ Thioredoxin, interaction of apoptosis signal-regulating kinase 1 and thioredoxin	[167]
Quercetin	tMCAO	Pre: 10 mg/kg 30 min before	↓ Infarct volume, neurological deficit ↑ Protein phosphatase 2A	[166]
Quercetin	tMCAO	Post: 10, 30, 50 mg/kg at the onset of reperfusion	↓ BBB injury, ROS, infarct volume, neurological deficit	[170]
Quercetin	pMCAO	Pre: 10 mg/kg 1 h before	↓ Intracellular calcium overload, glutamate excitotoxicity, caspase-3.	[168]
Myricetin	tMCAO	Pre: 20 mg/kg 2 h before and daily for 2 days after ischemia Pre: 25 mg/kg daily for 7 days	↓ Oxidative stress, apoptosis, neuronal loss, inflammation, infarct volume, ROS, neurological deficits ↑ Antioxidant enzymes, mitochondrial function, Nrf2 nuclear translocation, HO-1 expression	[171]
Myricetin	pMCAO	Pre: 1 mg/kg, 5 mg/kg, 25 mg/kg for 7 days	↓ IL-1β, IL-6, TNF-α, MDA, p38 MAPK, NF-κB/p65, apoptosis, infarct area, neurological deficit ↑ GSH/GSSG ratio, SOD, phosphorylated AKT	[174]
Myricetin	tMCAO	Pre: 25 mg/kg for 7 days	↓ Excitotoxicity, oxidative stress, inflammation, apoptosis	[173]
Kaempferol	tMCAO	Pre: 10,15 μmol/l 30 min before and immediately after ischemia Post: 7.5, 10 mg/kg single dose Post: 25, 50, 100 mg/kg daily for 7 days	↓ Metalloproteinase, anti-laminin staining, nitrosative-oxidative stress, caspase-9, apoptosis, poly-(ADP-ribose) polymerase, amyloid protein precursor, glial fibrillary acidic protein, phosphorylated STAT3, NF-κB p65, nuclear content of NF-κB p65, tumor necrosis factor α, interleukin 1β, intercellular adhesion molecule 1, matrix metallopeptidase 9, inducible nitric oxide synthase, myeloperoxidase, neuroinflammation, BBB injury, microglia activity, brain injury, neurological deficits	[175,176,181]
Naringenin	pMACO	Pre: 100 mg/kg daily for 4 days	↓ Neuroinflammation, edema, NOD2, RIP2, NF-κB, MMP-9, BBB injury, infarct volume, neurological deficits ↑ Claudin-5	[179]
Naringenin	tMCAO	Pre: 50 mg/kg daily for 21 days Post: 80 µM single dose	↓ Apoptosis, oxidative stress, edema, NF-κB, myeloperoxidase, nitric oxide, cytokines, neuroinflammation, glial activation, injury volume, neurological deficits ↑ Cortical neurons proliferation	[178,180]
Luteolin	tMCAO	Post: 20, 40, 80 mg/kg 0 and 12 h after ischemia	↓ Injury volume, edema, IL-1β, TNF-α, iNOS, COX-2, NF-κB, inflammation, neurological deficits ↑ Nrf2, PPARγ.	[181]
Luteolin	tMCAO	Post: 5, 10, 25 mg/kg single dose	↓ Oxidative stress, apoptosis, mRNA and protein of MMP9, infarct volume, neurological deficits ↑ PI3K/Akt	[182]
Luteolin	pMCAO	Post: 10, 25 mg/kg single dose post-ischemia	↓ MDA, Bax, oxidative stress, apoptosis, edema, infarct volume, neurological deficits ↑ SOD1, CAT, Bcl-2, claudin-5	[183]
Luteolin	pMCAO	Post: 5, 10 mg/kg 0 h and daily for 3 days survival	↓ Brain edema, TLR4, TLR5, p-p38, NF-κB infarct size, neurological deficit ↑Phospho-ERK	[184]
		Phenolic acids		
Caffeic acid	tMCAO	Pre: 10, 50 mg/kg 30 min before, 0, 1, 2 h, and every 12 h for 4 days after ischemia Pre: 0.1, 1, 10 µg/kg 15 min before, single dose	↓ Neuroinflammation, leukotrienes, neuron loss, 5-lipoxygenase, astrocyte proliferation, infarct volume, brain atrophy, neurological dysfunction ↑ NO	[185,186]
Caffeic acid	pMCAO	Post: 10 µmol/kg daily for 7 days	↓ MDA, CAT, XO, oxidative stress, lipid peroxidation, infarct size, neurological deficits ↑ GSH, NO	[187]
Caffeic acid	Global ischemia	Post: 10, 30, 50 mg/kg single dose	↓ Hippocampus injury, NF-κBp65, MDA, 5-LO, oxidative stress, memory deficits ↑ SOD	[189]
Caffeic acid	tMCAO	Post: 3, 10, 30 mg/kg 0, 2 h after ischemia	↓ MMP-2, MMP-9, edema, damage in penumbra, infarct volume, sensory-motor deficits, behavioral deficits	[188]
Ferulic acid	Global ischemia	Post: 28, 56, 112 mg/kg daily for 5 days	↓ Oxidative stress, mRNA caspase 3, mRNA Bax, hippocampus apoptosis, memory impairment ↑ mRNA Bcl-2, SOD	[195]
Ferulic acid	tMCAO	Post: 50, 100, 200 mg/kg daily for 7 days	↓ Hippocampus injury, neurological deficits ↑ In hippocampus, EPO and granulocyte colony-stimulating factor	[193]
Ferulic acid	tMCAO	Post: 100 mg/kg 0 h post-ischemia Post: 100 mg/kg 2 h post-ischemia Post: 100 mg/kg 24 h Pre: 100 mg/kg 24 h before ischemia Pre: 100 mg/kg 2 h before ischemia	Pretreatment 2 h before ischemia and posttreatment 2 h after ischemia ↓ Bax, astrocytosis, infarction volume	[194]
Ferulic acid	tMCAO	Pre: 80, 100 mg/kg Post: 100 mg/kg 30 min after ischemia	↓ Superoxide radicals, ICAM-1, NF-κB, infarct size, neurological deficits	[190]
Ferulic acid	tMCAO	Post; 100 mg/kg 0 h after ischemia	↓ ICAM-1 mRNA, Mac-1 mRNA, Mac-1, 4-HNE, 8-OHdG positive cells, TUNEL positive cells, caspase 3, microglia activity, apoptosis, macrophages, oxidative stress, inflammation	[191]
Ferulic acid	tMCAO	Post: 100 mg/kg 0 h, or 30 min or 2 h after ischemia	↓ PSD-95, nNOS, iNOS, nitrotyrosine, caspase-3, apoptosis, Bax, cytochrome c, MAP kinase ↑ Gamma-aminobutyric acid type B receptor, therapeutic window	[192]
P-coumaric acid	pMCAO	Post: 100 mg/kg single dose	↓ Oxidative damage, MDA, apoptosis, caspase-3, caspase-9, edema, infarct volume, neurological deficits ↑ SOD, NRF-1	[196]
P-coumaric acid	Global ischemia	Pre: 100 mg/kg for 2 weeks before ischemia	↓ MDA, oxidative stress, hippocampal neuronal death, infarct volume, brain damage ↑ Catalase, superoxide dismutase	[197]
Chlorogenic acid	tMCAO	Post: 3, 10, 30 mg/kg 0, 2 h after ischemia Pre: 15, 30, 60 mg/kg for 1 week	↓ BBB, oxidative stress, MMP-2, MMP-9, edema, infarct volume, sensory-motor deficits, behavioral deficits	[198,199]
Chlorogenic acid	tMCAO	Post: 30 mg/kg 2 h after ischemia	↓ Cytochrome c, caspase-3, cleaved caspase-3, neurological deficits ↑ Phospho-PDK1, phospho-Akt, phospho-Bad	[202]
Chlorogenic acid	tMCAO	Pre: 15, 30, 60 mg/kg once a day for 1 week	↓ Mortality, infarction area, injury of hippocampus, cortex lesions, neurological deficit ↑ EPO, HIF-1α, NGF	[199]
Chlorogenic acid	Repeated global ischemia	Post: 20, 100, 500 mg/kg single dose	↓ Oxidative stress, apoptosis, MMPs, infarct volume, memory deficits ↑ SOD, GSH	[200]
Chlorogenic acid	Embolic strokes with rtPA	Post: 50 mg/kg 5 min, 1, 3 h after ischemia	↓ Behavioral deficits ↑ Therapeutic window	[205]
Chlorogenic acid	tMCAO	Post: 30 mg/kg 2 h after ischemia	↓ TUNEL-positive cells, caspase-3 and -7, oxidative stress, edema, infarct size, neurological damage	[201]
Chlorogenic acid	tMCAO	Post: 30 mg/kg 2 h post-ischemia	↓ Reactive oxygen species, oxidative stress, NF-κB, IL-1β, TNF-α, microglia, astrocyte activation, inflammation, cortex pathology	[204]
Chlorogenic acid	tMCAO	Post: 30 mg/kg/d 3 days after ischemia	↓ Cerebral cortex apoptosis, infarct volume ↑ Angiogenesis, VEGFA, PI3K/Akt signaling	[202]
Gallic acid	tMCAO	Pre: 50 mg/kg daily for 7 days Pre: 50 mg/kg single dose	↓ Oxidative stress, apoptosis, neuroinflammation, mitochondrial dysfunction, injury size, neurological deficits	[207,209]
Gallic acid	Global ischemia	Pre: 100 mg/kg/d for 10 days	↓ BBB injury, MDA, oxidative stress, hippocampus EEG changes, anxiety, behavioral deficits	[210]
Gallic acid	Global ischemia	Post: 25, 50 mg/kg/d for 1 week	↓ Oxidative stress, depressive symptoms	[208]
Ellagic acid	Photothrombotic model	Pre: 10, 30 mg/kg 24 h before and 0 h post-ischemia	↓ Apoptotic cells, infarct size, neurological deficits	[206]

P2VO—permanent occlusion of both common carotid arteries, tMCAO—transient middle cerebral artery occlusion, pMCAO—permanent middle cerebral artery occlusion, BBB—blood–brain barrier, ROS—reactive oxygen species, Nrf2—nuclear factor erythroid 2-related factor 2, HO-1—heme oxygenase 1, STAT3—signal transducer and activator of transcription 3, NF-κB p65—nuclear-kappa B factor p65, NOD2—nucleotide-binding oligomerization domain containing 2, RIP2—receptor-interacting-serine/threonine-protein kinase 2, MMP-9—matrix metallopeptidase-9, IL-1β—interleukin-1 beta, IL-6—interleukin-6, MDA—malondialdehyde, p38 MAPK—p38 mitogen activated protein kinase, GSH/GSSG—glutathione/glutathione disulfide, SOD—superoxide dismutase, AKT—protein kinase B, TNF-α—tumor necrosis factor-α, MAPKs—mitogen-activated protein kinases, ERK—extracellular signal-regulated kinase, iNOS—inducible nitric oxide synthase, COX-2—cyclooxygenase-2, PPARγ—peroxisome proliferator-activated receptor gamma, PI3K—phosphatidylinositol 3-kinase, CAT—catalase, TLR4—toll-like receptor 4, p-p38—phospho-p38, XO—xanthine oxidase, NO—nitric oxide, 5-LO—5-lipoxygenase, EPO—erythropoietin, ICAM-1—intercellular adhesion molecule 1, NRF1—nuclear respiratory factor 1, VEGFA—vascular endothelial growth factor A, EEG—electroencephalography. ↓—decrease, ↑—increase.

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
