# Peer review of "Apitherapy in Post-Ischemic Brain Neurodegeneration of Alzheimer’s Disease Proteinopathy: Focus on Honey and Its Flavonoids and Phenolic Acids"

_molecules, 2023, doi:10.3390/molecules28155624_

Round 1

Reviewer 1 Report

The review MS entitled “Apitherapy in post-ischemic brain neurodegeneration of Alzheimer's disease proteinopathyA23R365: focus on honey and its flavonoids and phenolic acids” was reviewed.

The authors have gathered information related to the importance of honey chemical constituents such as flavonoids etc. and their possible therapeutic use in combating brain ischemia. Moreover, they also summarized various classes of metabolites occur in honey. The article is impressive and suitable. Some of my suggestions are:

1.      Add graphical abstract in the MS.

2.      The abstract section of the review lacking explanation about the possible therapeutic role of phenolic acids and flavonoids, the author should give a slight insight into its significances in the abstract section.

3.      Introduction; line 79; put ‘full stop’ after references [9, 19] to complete the sentence.

4.      Line 116; remove space after reference [43].

5.      Figure 1 and 2 should be modified and the underlining should be removed (correct the spelling/grammar mistakes). Put the figures in BMP or JPG format.

6.      This review lacks a proper mechanistic diagrammatic presentation of ischemic stroke and neurodegeneration how that happens. The author should provide it in the revised version.

7.      The Alzheimer’s disease and dementia is discussed with not much detail. A detail background of Alzheimer’s disease should be supported by an appropriate reference as for guidance; https://doi.org/10.3390/molecules27082468, doi10.3390/molecules26237168; doi10.1097/CM9.0000000000001706, etc.

8.      There should be a figure indicating the relationships of various causative factors such as tau protein, amyloid-β, oxidative stress etc with that of neuronal degeneration and ischemic stroke.

9.      There is very little description of phenolic acids in this review, the author should look insight it more.

10.   The conclusion section is very much lengthy; author should rewrite the conclusion in a simple and concise manner highlighting the keys aspects.

11.   The author has used dark colored legends in Figure 2 and not in a uniform sequence with some underline legends, in my opinion it should be redesigned to make it more explanatory.

12.   A sunburst diagram should be added that could summarize the active constituents, their source within honey and the particular target of therapy in brain ischemia. This will increase a comprehensiveness in this study.

13.   If possible, any SAR studies reported should also be included in order to check the actual molecular basis of interactions of various compounds with ischemia targets.

14.   Why did the authors not explain any Biomarkers for these diseases?

15.   Also, the current medications and causes of their low effectiveness should also be the part of introduction.

16.   There are many flaws found in the review which should be carefully addressed before publishing.

17.   In my opinion this review has much pronounced future perspectives; however, many points as well as typos error should be addressed before publishing it.  

 Extensive editing of English language required

Author Response

The review MS entitled “Apitherapy in post-ischemic brain neurodegeneration of Alzheimer's disease proteinopathyA23R365: focus on honey and its flavonoids and phenolic acids” was reviewed.

The authors have gathered information related to the importance of honey chemical constituents such as flavonoids etc. and their possible therapeutic use in combating brain ischemia. Moreover, they also summarized various classes of metabolites occur in honey. The article is impressive and suitable.

Thanks.

Some of my suggestions are:

  1. Add graphical abstract in the MS.-Done.
  2. The abstract section of the review lacking explanation about the possible therapeutic role of phenolic acids and flavonoids, the author should give a slight insight into its significances in the abstract section.-Done.
  3. Introduction; line 79; put ‘full stop’ after references [9, 19] to complete the sentence.-Done.
  4. Line 116; remove space after reference [43].-Done.
  5. Figure 1 and 2 should be modified and the underlining should be removed (correct the spelling/grammar mistakes). Put the figures in BMP or JPG format.-Done.
  6. This review lacks a proper mechanistic diagrammatic presentation of ischemic stroke and neurodegeneration how that happens. The author should provide it in the revised version.- Ischemic mechanisms are described in detail in section 6. Adding more graphs will result in duplicated data.
  7. The Alzheimer’s disease and dementia is discussed with not much detail. A detail background of Alzheimer’s disease should be supported by an appropriate reference as for guidance; https://doi.org/10.3390/molecules27082468, doi: 10.3390/molecules26237168; doi10.1097/CM9.0000000000001706, etc.- The aim of the review is not Alzheimer's disease itself, but post-ischemic pathology identical to Alzheimer's disease. We included 2 papers suggested by the reviewer in the manuscript. However, the third paper on epigenetic elements in Alzheimer's disease was not included because it is completely unrelated to the subject of the review.
  8. There should be a figure indicating the relationships of various causative factors such as tau protein, amyloid-β, oxidative stress etc with that of neuronal degeneration and ischemic stroke.

      -Proposing another figure on the effect of tau protein, amyloid-β, oxidative stress, etc. on neurons and ischemic stroke is another proposal for duplicating the data that is in section 6.

  1. There is very little description of phenolic acids in this review, the author should look insight it more. – There are 27 references in table 1 detailing the administration and details - data on the mechanisms of action, at the moment more data is simply not available. In addition, additional descriptions are on page 13 line 500-530.
  2. The conclusion section is very much lengthy; author should rewrite the conclusion in a simple and concise manner highlighting the keys aspects. - Conclusion shortened by about 20 lines.
  3. The author has used dark colored legends in Figure 2 and not in a uniform sequence with some underline legends, in my opinion it should be redesigned to make it more explanatory. - We think the colors are contrasting. Now the figure is presented in JPEG, the typos have been corrected and the arrows are described under the figure.
  4. A sunburst diagram should be added that could summarize the active constituents, their source within honey and the particular target of therapy in brain ischemia. This will increase a comprehensiveness in this study. - We do not share this proposal, in our opinion it would be a duplication of the data from table 1.
  5. If possible, any SAR studies reported should also be included in order to check the actual molecular basis of interactions of various compounds with ischemia targets. - There is currently no data available on this subject.
  6. Why did the authors not explain any Biomarkers for these diseases? - This issue was not the topic of the MS.
  7. Also, the current medications and causes of their low effectiveness should also be the part of introduction. - These details can be actually found in the introduction.
  8. There are many flaws found in the review which should be carefully addressed before publishing. - Done.
  9. In my opinion this review has much pronounced future perspectives; however, many points as well as typos error should be addressed before publishing it. - Thanks for the comments: done and corrected.

All changes in MS are in red.

Reviewer 2 Report

In this manuscript entitled “Apitherapy in post-ischemic brain neurodegeneration of Alzheimer's disease proteinopathy: focus on honey and its flavonoids and phenolic acids” the authors have reviewed the literature to document the neuroprotective role of honey and its antioxidant components, flavonoids and phenolic acids in post-ischemic brain neurodegeneration. Overall, the authors have comprehensively covered the available literature and provided a detailed function of individual components of honey in neuroprotection. I have the following comments:

1-    On page 12 line 574, the authors have cited two papers from the same group that administered honey directly and show its effect on neuroprotection. Can the authors clarify which model of brain ischemia was used and how the honey was administered e.g the route of honey administration.

2-    Along the same line, unless I missed it, are there any other paper/papers from a different group that showed neuroprotective function of honey? If yes, please include those as well on page 12, line 574.      

3-    Minor points: the sentence 244-245 seems incomplete, please correct it.

4- In the fig2 there is a typo Phenilic acid

Minor grammatical mistakes

Author Response

In this manuscript entitled “Apitherapy in post-ischemic brain neurodegeneration of Alzheimer's disease proteinopathy: focus on honey and its flavonoids and phenolic acids” the authors have reviewed the literature to document the neuroprotective role of honey and its antioxidant components, flavonoids and phenolic acids in post-ischemic brain neurodegeneration. Overall, the authors have comprehensively covered the available literature and provided a detailed function of individual components of honey in neuroprotection.

Thanks.

I have the following comments:

  • On page 12 line 574, the authors have cited two papers from the same group that administered honey directly and show its effect on neuroprotection. Can the authors clarify which model of brain ischemia was used and how the honey was administered e.g the route of honey administration. – Route p.o. Other explanations are in table 1 in the first row. Model p2VO-permanent 2 vessel occlusion.

2-    Along the same line, unless I missed it, are there any other paper/papers from a different group that showed neuroprotective function of honey? If yes, please include those as well on page 12, line 574.   - We have not found any more papers of this type.

3-    Minor points: the sentence 244-245 seems incomplete, please correct it. Done.

4- In the fig2 there is a typo Phenilic acid. - Thank you. Corrected.

All changes in MS are in red.

Round 2

Reviewer 1 Report

The suggestions were incorporated and the quality of manuscript is improved. I recommend this manuscript for publication in molecules.

Ok